# Enhanced Diagnostics for Corneal Ectatic Diseases: The Whats, the Whys, and the Hows

**DOI:** 10.3390/diagnostics12123027

**Published:** 2022-12-02

**Authors:** Louise Pellegrino Gomes Esporcatte, Marcella Q. Salomão, Alexandre Batista da Costa Neto, Aydano P. Machado, Bernardo T. Lopes, Renato Ambrósio

**Affiliations:** 1Department of Ophthalmology, Federal University of São Paulo, São Paulo 04023, Brazil; 2Rio de Janeiro Corneal Tomography and Biomechanics Study Group, Rio de Janeiro 20520-050, Brazil; 3Instituto de Olhos Renato Ambrósio, Rio de Janeiro 20520-050, Brazil; 4Brazilian Artificial Intelligence Networking in Medicine-BrAIN, Rio de Janeiro 20520-050, Brazil; 5Instituto Benjamin Constant, Rio de Janeiro 22290-255, Brazil; 6Belo Horizonte Evangelical Hospital, Betin Unit, Belo Horizonte 30220-330, Brazil; 7Computing Institute, Federal University of Alagoas, Maceió 57072-900, Brazil; 8School of Engineering, University of Liverpool, Liverpool L69 3GH, UK; 9Department of Ophthalmology, Federal University the State of Rio de Janeiro (UNIRIO), Rio de Janeiro 22290-240, Brazil

**Keywords:** keratoconus, corneal ectasia, multimodal corneal imaging, corneal biomechanics, genetics

## Abstract

There are different fundamental diagnostic strategies for patients with ectatic corneal diseases (ECDs): screening, confirmation of the diagnosis, classification of the type of ECD, severity staging, prognostic assessment, and clinical follow-up. The conscious application of such strategies enables individualized treatments. The need for improved diagnostics of ECD is related to the advent of therapeutic refractive procedures that are considered prior to keratoplasty. Among such less invasive procedures, we include corneal crosslinking, customized ablations, and intracorneal ring segment implantation. Besides the paradigm shift in managing patients with ECD, enhancing the sensitivity to detect very mild forms of disease, and characterizing the inherent susceptibility for ectasia progression, became relevant for identifying patients at higher risk for progressive iatrogenic ectasia after laser vision correction (LVC). Moreover, the hypothesis that mild keratoconus is a risk factor for delivering a baby with Down’s syndrome potentially augments the relevance of the diagnostics of ECD. Multimodal refractive imaging involves different technologies, including Placido-disk corneal topography, Scheimpflug 3-D tomography, segmental or layered tomography with layered epithelial thickness using OCT (optical coherence tomography), and digital very high-frequency ultrasound (VHF-US), and ocular wavefront. Corneal biomechanical assessments and genetic and molecular biology tests have translated to clinical measurements. Artificial intelligence allows for the integration of a plethora of clinical data and has proven its relevance in facilitating clinical decisions, allowing personalized or individualized treatments.

## 1. Introduction

Keratoconus (KC) is the most common ectatic corneal disease (ECD), which comprises a class of disorders characterized by progressive thinning and subsequent bulging of the cornea, causing irregular astigmatism [1,2]. When evaluating patients with ectatic disorders of the cornea, the clinician must consider the different fundamental diagnostic strategies (Table 1): screening, confirmation of the diagnosis, classification of the type of ectasia, staging severity, prognostic evaluation, and clinical follow-up for individualized treatments. Refractive surgery boosted extensive developments in diagnostic and therapeutic technologies for ECD [3,4,5,6]. Therapeutic refractive procedures are less invasive alternatives to keratoplasty. Corneal crosslinking, customized ablations, and intracorneal ring segments (ICRSs) can be indicated for such patients. Introducing such options for managing ECD determined the need for an improved disease evaluation.

Besides the paradigm shift in managing such patients, enhancing the sensitivity to detect very mild forms of ECD is relevant for identifying patients at higher risk for progressive iatrogenic ectasia after laser vision correction (LVC). The current concept for assessing ectasia risk before LVC combines the characterization of corneal properties, related to the inherent susceptibility for ectasia progression, and the impact of the LVC procedure on the corneal structure. External mechanical factors, such as eye rubbing and pressurizing the eyes during sleep, also play a significant role. This concept contemplates the two-hit hypothesis for the pathogenesis of ECD [7].

Multimodal refractive imaging involves different technologies, including Placido-disk corneal topography, Scheimpflug 3-D tomography, segmental or layered tomography with layered epithelial thickness using OCT (optical coherence tomography), digital very high-frequency ultrasound (VHF-US), and ocular wavefront. Corneal biomechanical assessments were translated from mathematical models. In vitro laboratory destructive tests to clinically measure beyond shape analysis have been promising as a crucial tool for enhancing the accuracy of identifying mild forms of ECD and the capacity to characterize ectasia susceptibility [7,8].

Artificial intelligence (AI) has proven its relevance in integrating the overabundance of data generated for facilitating clinical decisions. Genetic and molecular biology tests are promising to further enhance diagnostic accuracy for allowing personalized treatments. Moreover, the hypothesis that mild or fruste keratoconus is a risk factor for a mother delivering a baby with Down’s syndrome opened a new horizon for the relevance of the diagnostics of ECD [9]. This article presents a prospective review of the diagnostic methodology for ECD.

## 2. Multimodal Imaging

The concept of multimodal corneal imaging was announced to integrate several diagnostic tools offered for corneal and anterior segment imaging (Table 2) [5]. Placido disk-based corneal topography improved the ability to detect abnormalities in mild corneal ectasia, even in patients with normal distance-corrected visual acuity and slit-lamp examinations within normality [4,10,11].

Anterior segment tomography with 3D reconstruction of the cornea offered more detail about corneal architecture with a range of quantitative indices derived from the front and back elevation along with pachymetric maps [5,12,13,14]. Different studies involving eyes with typically “innocent” topography, from patients with clinical ectasia identified in the fellow eye, confirmed the capacity of corneal tomography to improve the accuracy of detecting mild or subclinical ectatic disease [4,15,16,17,18,19,20]. Corneal tomographic parameters revealed an excellent ability to identify susceptibility to developing ectasia after LASIK (Laser-Assisted in Situ Keratomileusis) in retrospective studies involving patients with such complications [13,21,22]. The need to go beyond corneal shape evaluation for describing ectasia risk within the biomechanical domain was supported and promoted [23,24].

## 3. Screening for Ectasia Risk before Laser Vision Correction

The key to refractive surgery screening is to identify cases with mild ectasia and cases of high susceptibility or predisposition for biomechanical failure and ectasia after LVC [25,26]. Corneal ectasia has been a severe complication of LASIK surgery since the first report by Seiler in 1998 [27]. Clinical risk factors associated with ectasia include preoperative ectactic corneal disease, young age, and low preoperative pachymetry [21,28,29,30].

Corneal ectasia is the consequence of a biomechanical decompensation of the stroma. This could be related to either the impact of the procedure on the corneal structure or the preoperative individual biomechanical properties. The present understanding is that a combination of these factors determines stability or progression of ectasia after LVC [7,8,31]. However, some cases with low preoperative risk factors can develop ectasia, whereas others with high probabilities of developing ectasia continue to be stable [7]. Long-term stability after LVC is determined by the preoperative biomechanical strength of the patient’s corneal stroma, the amount of biomechanical alteration caused by the surgery, and the postoperative stress load on the cornea [31].

### 3.1. Corneal Topography

Placido disk-based corneal topography projects a sequence of concentric rings onto the anterior corneal surface and uses quantitative data to generate color-coded maps [32,33]. For the diagnosis of KC, one of the most commonly used indices is Rabinowitz and McDonnell’s [34]. Corneal topography epitomizes a revolution in corneal imaging. It has the sensitivity to detect ectatic disease before any loss of best-corrected visual acuity, and any significant slit-lamp examination findings acquired [10,11]. Consequently, corneal topography is considered an essential examination when screening refractive surgery candidates [35]. Randleman and co-workers developed the Ectasia Risk Scoring System with corneal topography, pachymetric measurements, and clinical variables [29,36].

However, limitations of topography examination were documented after literature presented patients who developed post-LVC ectasia, despite normal preoperative anterior surface maps [7,28,37] and patients with renowned risk factors and abnormal topographic maps who stayed stable years after LVC [38]. Therefore, there is a need for a complete characterization of the cornea to improve screening for ectasia susceptibility of refractive candidates.

### 3.2. Corneal Tomography

The Pentacam (Oculus, Wetzlar, Germany) uses a rotating Scheimpflug camera and a frontal view illumination system to reconstruct topographic images of the cornea and anterior segment, and diverse indices have been proposed to increase the diagnosis of KC.

The Pentacam Belin–Ambrósio Enhanced Ectasia Display (BAD) was designed as a clinical tool to assist clinical diagnosis of KC and ECD [15,16,17,19,39]. Tomographic parameters are displayed as the standard deviation from normality to disease (d values), including front and back elevation at the thinnest point, change in anterior and posterior elevation of the standard and enhanced best-fit sphere (BFS), the thinnest value and its vertical location, pachymetric progression index, Ambrósio’s relational thickness, and maximal curvature (KMax). A final ‘D’ value is calculated based on linear regression analysis, which weights each parameter differently [8,14,15].

Lopes and collaborators proposed using random forest (RF) to enhance pattern recognition [40]. The Pentacam Random Forest Index (PRFI) was developed to detect ectatic corneal disease among the following groups, performing with better accuracy than any individual tomographic parameter: normal eyes (stable LASIK cases), clinical KC, normal topographic eyes with very asymmetric ectasia (VAE-NT), and eyes with higher ectasia susceptibility (preoperative data of post LASIK ectasia). Compared with the BAD-D, which correctly classified only 55.3% of the post-LASIK ectasia, the PRFI correctly detected 80%. Considering the recognition of the VAE-NT, the PRFI presented a sensitivity of 85.2% and specificity of 96.6% in the independent test set [40].

### 3.3. Segmental Corneal Tomography

Corneal epithelial indices for identifying KC were developed with this technology, and studies proposed this approach as a valuable instrument in identifying milder forms of the disease [41,42]. Working with optical coherence tomography (OCT) technology, Huang and collaborators developed a similar approach with an extended epithelial thickness map and different indices to detect KC in its initial stages [43,44]. Sinha-Roy and coauthors also investigated the irregularity of the Bowman’s layer in normal and ectatic corneas and proposed a new Bowman’s roughness index. This index had a good performance in detecting KC and, when combined with the BAD-D and epithelial thickness data, improved the sensitivity for detecting mild forms of ectasia [45].

Studies also demonstrated the use of this technology to investigate flap thickness reproducibility, to understand corneal refractive surgery complications, and to measure epithelial changes after refractive surgery [46].

## 4. Corneal Biomechanical Assessment

The biomechanical analysis first gained relevance in refractive surgery to recognize patients at higher risk of developing iatrogenic ectasia after LVC. In addition, biomechanical customization can further improve the probability and efficacy of these elective procedures [6,47,48,49]. Studies demonstrated the capacity of this technology to identify mild, “forme fruste” or subclinical KC in eyes with a normal anterior curvature (topography) from patients with clinical ectasia in the fellow eye (very asymmetric ectasia—VAE) [15,16].

The Corvis ST analyzes corneal deformation parameters based on the dynamic examination of the corneal response. The artificial intelligence (AI) algorithms confirmed that the arrangement of deformation parameters improved the accuracy of discriminating healthy and KC eyes, even in mild stages [50]. Furthermore, the Corvis ST waveform analysis of the deformation amplitude and deflection amplitude signals offered outstanding performance in distinguishing normal, suspect, and KC eyes [51].

In 2014, two main parameters were developed by a multicentric international investigation group for enhancing ectasia detection, the Corvis Biomechanical Index (CBI) and the tomographic biomechanical index (TBI) [4,52,53]. The TBI was established with a random forest model using data from the corneal deformation response and the corneal tomography to enhance the capability to divide normal and altered eyes. Vinciguerra and co-workers demonstrated in the training dataset that, with a cutoff value of 0.5, CBI correctly identified 98.2% of KC cases among normal with 100% specificity and 94.1% sensitivity AUC of 0.983. Later, in the validation dataset, the same cutoff value classified 98.8% of cases, with 98.4% specificity and 100% sensitivity and an AUC of 0.999 [54]. The cutoff of 0.79 provided 100% sensitivity and specificity to detect clinical ectasia formed by KC and very asymmetric ectasia (VAE-E) cases. For the eyes with a normal topographic standard, an optimized cutoff of 0.29 offered 90.4% sensitivity and a specificity of 96%, with an area under the ROC curve of 0.985. The AUC of the TBI was statistically higher than all other analyzed parameters, including the CBI [48].

Next, a study determined that the TBI was the most sensitive index to verify mild ectasia [48]. Subsequent external validation studies validated that the TBI could detect mild forms of ectasia in very asymmetric ectasia with typical topography (VAE-NT) cases [47,55,56,57,58]. Some of these studies found a comparatively lower sensitivity for the VAE-NT eyes (some with typical topography and tomography—NTT). However, it is pertinent to note that various of these cases could be genuinely unilateral ectasia due to mechanical trauma [30,59]. Recently, a novel optimized version of the TBI (TBIv2) was developed with significantly higher accuracy (0.945) for detecting VAE-NT (84.4% sensitivity and 90.1% specificity; cutoff 0.43) and similar AUC for clinical ectasia (0.999; 98.7% sensitivity; 99.2% specificity; cutoff 0.8). Considering all cases, the TBIv2 had a higher AUC (0.985) than TBIv1 (0.974) [Ambrósio et al., data in press 2022].

## 5. Classification of Ectatic Disease

The ECD includes a group of diseases characterized by progressive thinning, followed by protruding, of the corneal structure, involving keratoglobus, pellucid marginal degeneration (PMD), and KC [1].

Keratoglobus typically appears bilaterally and is characterized by a generalized thinning and rounded protrusion of the entire cornea, causing an irregular corneal topography with increased corneal fragility due to extreme thinning. It is a congenital disorder and commonly related to connective tissue diseases; though, recent reports suggest that keratoglobus may also be developed and associated with blepharitis, vernal keratoconjunctivitis atopy, dysthyroid eye disease, corneal traumas, and extreme eye rubbing [54].

PMD is described by a typical narrow band of corneal thinning close to the limbus but conserving an area of 1–2 mm. It is undefined whether these are distinctive phenotypic variants of KC or, in fact, distinct disorders [55].

Different types of corneal refractive surgery may be related with iatrogenic progressive corneal ectasia, like after LASIK in patients with altered biomechanical properties, deemed as forme fruste keratoconus (FFKC) [27].

KC is the most commonly described clinical condition. It is bilateral, asymmetric, and usually a progressive ectatic corneal disease, characterized by chronic biomechanical failure and stromal thinning [1,56]. The whole pathophysiology of ECD is not understood. However, there is an agreement that an interaction between genetic and environmental factors activates the disease, although the exact role of each of these factors can be variable. As there is no conclusive genetic test for KC, several cases denoted as KC may be associated with a secondary cause. In such cases, a diversity of risk factors, for example, contact lens wear, eye rubbing, and allergic disease, may be associated with the pathogenesis.

The two-hit hypothesis proposed by McGhee [57] can be corroborated by case reports that show twins with different degrees of KC involvement. For example, two identical 48-year-old female twins, one of whom had rubbed one of her eyes during early adulthood, had very asymmetric ectasia with normal anterior curvature and topography in one eye, and the other twin, who had not rubbed either of her eyes, had normal topography in both eyes [58]. Another case concerned 16 years old fraternal twins, one of whom was observed to have asymmetric KC with advanced KC in the right eye (OD) and mild in the left eye (OS). In contrast, his brother had an FFKC in both eyes. Both twins had an ocular allergy, but the one with asymmetric KC admitted to more eye rubbing than the other.

## 6. Genetics and Molecular Biology

The genetic description of KC is a true challenge. KC development has been associated with many genes, such as VSX 1, miR-184, DOCK9, SOD1, RAB3GAP1, and HGF [59]. The identification of at least 17 genomic loci in KC patients showed the genetic heterogeneity of the disease [60], complemented by the description of both autosomal dominant and recessive patterns [61].

Shortly, molecular biology might assist in the diagnosing and classifying of KC. Histopathologic studies described molecular and cellular changes related to the pathogenesis of KC, including extracellular matrix degeneration. This suggested an up-regulation of degradative enzymes, oxidative stress, and inflammation [62,63], which could eventually change the definition of the disease.

Adding genetic and molecular biological studies to the evaluation of KC is vital since studies detected that mothers with Down syndrome (DS) children are more likely to have KC and thinner, steeper, and softer corneas compared to mothers with normal children [9]. Another study reviewed the association between KC and DS, showing increasing evidence that supported the elevated risk (>100 times) of KC in DS patients. The genetic association of sequence variants within, or near the *COL6A1* and *COL6A2* genes on Chr21, with KC provided an additional functional link between KC and DS [64].

A recent study showed that Lactoferrin (LTF) and Toll-like Receptors 2 (TLR2) were clinically and molecularly interrelated, increasing knowledge about KC pathophysiology and opening the door to future therapies. The dysregulation of LTF and TLR2 in the ocular surface of KC patients contributed to KC severity by maintaining a detrimental chronic immune–inflammatory state. The regulation of these immunomodulatory properties may be a potential therapeutic approach for KC [65].

### 6.1. Follow-Up

#### 6.1.1. Belin’s ABC + D(DCVA) and the Corvis-Derived Parameter ‘E’

A novel biomechanical KC staging parameter ‘E’ [66], based on the Corvis Biomechanical Factor (CBiF), provides a measure for different stages of the biomechanical destabilization of the cornea [67] as an addition to tomographic ABCD ectasia/KC staging [68,69].

The ABCDE staging purpose is not to biomechanically diagnose KC based on these parameters but to increase the severity classification. The arrangement of tomographic and biomechanical parameters may offer clinical benefits over using either alone. Further clinical application of the enhanced staging system is necessary to determine its clinical applicability [66].

#### 6.1.2. Stiffness Parameter at First Applanation (SPA-1)

Some biomechanical parameters, such as the Stiffness Parameter at first Applanation (SPA-1), can help regarding the prognosis of the disease, as we observed in the case of a 13-year-old male patient with progressive ectasia in the OS and FFKC in the OD. The prediction of the high possibility of disease progression in OS could be suspected at the first examination, in which the SPA-1 was very low (25.8). Figure 1 shows the Pentacam differential map showing progression in OS (B-D). Observing only K max, we tended to believe that the KC improved from 60.6 in February 2021 to 59.5 in August 2021, but there was an increase in zonal curvature. At six months of follow-up, Figure 2 showed progression in all parameters of the Belin ABCD display in OS.

### 6.2. Prognostic

Patients and their families need to understand that surgery in KC is a therapeutic surgery and not a refractive procedure [70], and the principal purpose of surgical treatment is to reestablish vision.

However, the paradigm shift defined by Seiler encountered a paradox. The paradox is associated with the fact that, previously, visual rehabilitation was the primary purpose of KC treatment and was proposed after visual loss. Contrariwise, the purpose of newer treatment modalities is to prevent visual loss before it happens. Therefore, no surgery is indicated if it is not necessary, such as ECDs with good vision with glasses and no signs of progression, but is, nevertheless, done as soon as possible if vision declines or ECD progresses [71].

Whether to perform any procedure or not, and determining the most suitable strategy for each patient, when surgery is necessary, merits personalized attention. Patient compliance and advice against eye rubbing are essential, along with treating ocular allergy and optimizing ocular surface.

We reported a very asymmetric ectasia case with moderate keratoconus OD and “low K” mild keratoconus in OS. In Figure 3 (A and B), the Corvis-ST Biomechanical/Tomographic Assessment showed a SPA-1 which was a prognostic factor of progression in the intersection between normal and KC in OD. We observed, in OS topometric and tomographic stability (Figure 4), and slight alteration in the epithelial map, thin cornea without significant alteration in the epithelium (Figure 5). We need to point out that this epithelial map was limited to 6 mm, and new OCT devices could measure up to 9 or 10 mm.

We proceeded with ICRS implantation in OD, and Figure 6 shows the appearance of the ring in the biomicroscopy and its position with the OCT. The Biomechanical Comparison Display showed a stiffer cornea after the ICRS implantation (Figure 7).

## 7. Conclusions

Over recent decades, there has been ongoing concern about KC and ECD. This could permit us to consider KC as a clinical disorder and a distinct subspecialty in ophthalmology. Multimodal imaging may assist in comprehensively evaluating diagnosis, staging, prognosis, and treatment planning.

In refractive surgery, the key to screening is to recognize cases with mild ectasia and to distinguish each cornea in terms of its susceptibility to suffering biomechanical failure and ectasia [26]. The integration of tomographic and biomechanical data demonstrated the possibility of increasing the accuracy of identifying ectatic disease and detecting susceptibility to developing this complication after LVC [47,48,72]. Permanent research and advances in this field promote integrated multimodal corneal imaging, biomechanics, molecular biology, and genetics. Artificial Intelligence increases efficacy in patient care in this setting [6].

## Figures and Tables

**Figure 1 diagnostics-12-03027-f001:**
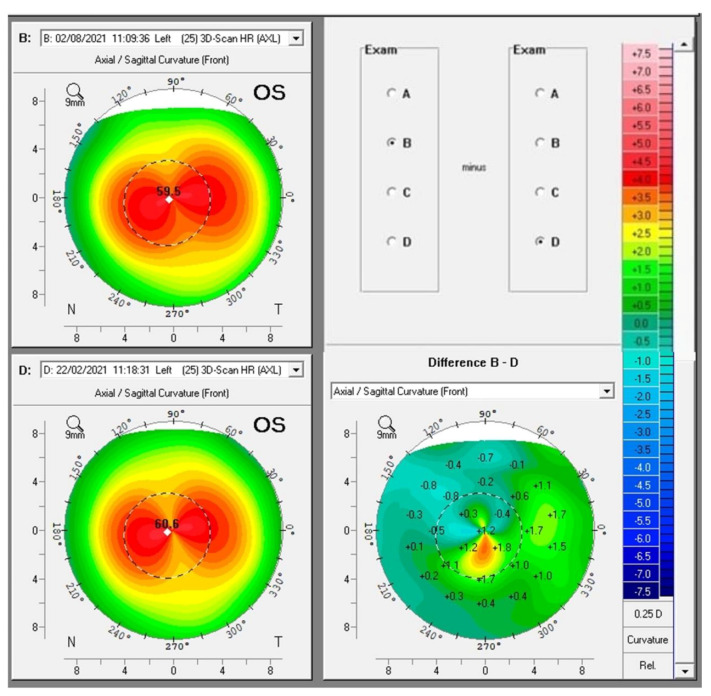
Pentacam differential map showing progression in OS (B-D). Observing only K max, we tended to believe that the KC improved from 60.6 in February 2021 to 59.5 in August 2021, but there was an increase in zonal curvature.

**Figure 2 diagnostics-12-03027-f002:**
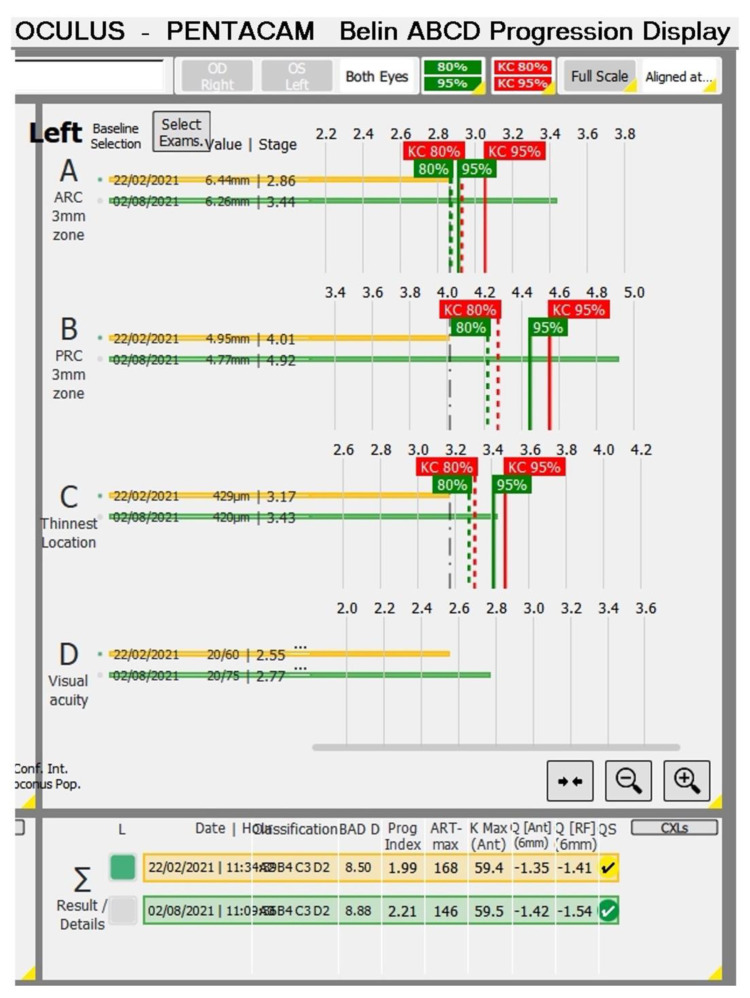
The Belin ABCD display shows progression in all parameters OS.

**Figure 3 diagnostics-12-03027-f003:**
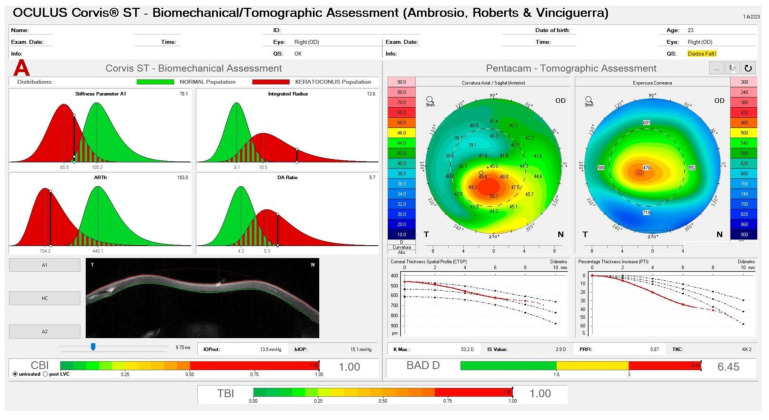
Corvis-ST Tomographic Biomechanical Display (ARV) from OD (**A**) showing a low SPA-1 (78.1). OS (**B**) which confirmed the diagnosis of VAE was by BAD-D of 2.29, CBI of 0.89, and the TBI of 1.0.

**Figure 4 diagnostics-12-03027-f004:**
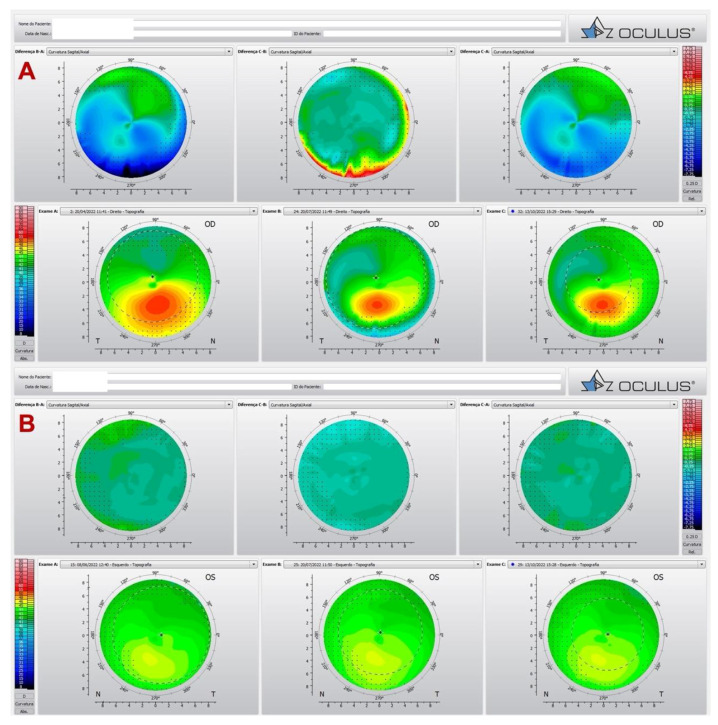
Pentacam differential map showing in OD (**A**) the topography before and after the ICRS, and in OS (**B**) topometric and tomographic stability along the time.

**Figure 5 diagnostics-12-03027-f005:**
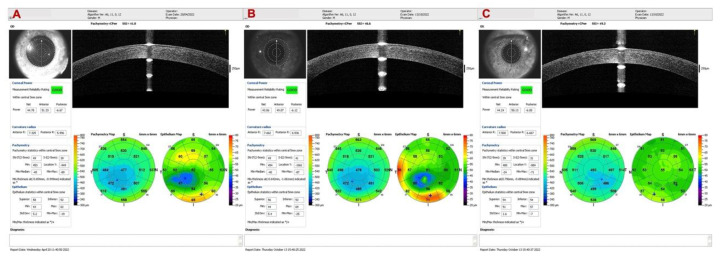
Epithelial maps of OD before (**A**) and after ICRS (**B**) and of OS (**C**). OS shows a slight alteration in the epithelial map, with a thin cornea without significant alteration in the epithelium.

**Figure 6 diagnostics-12-03027-f006:**
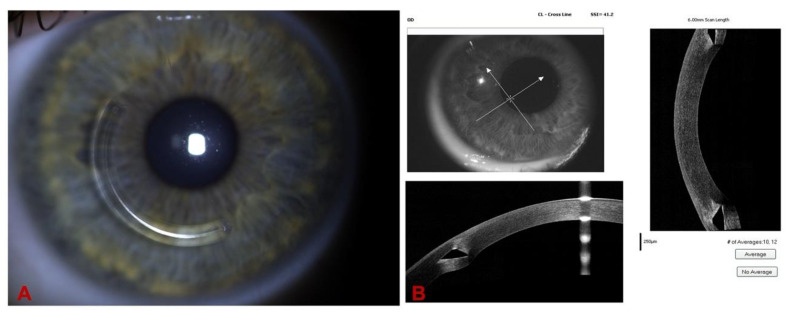
(**A**). Slit-lamp biomicroscopy of the Keraring AS (Mediphacos, Belo Horizonte, Brazil) ICRS and its position with the OCT in (**B**), demonstrating the progressive thickness profile.

**Figure 7 diagnostics-12-03027-f007:**
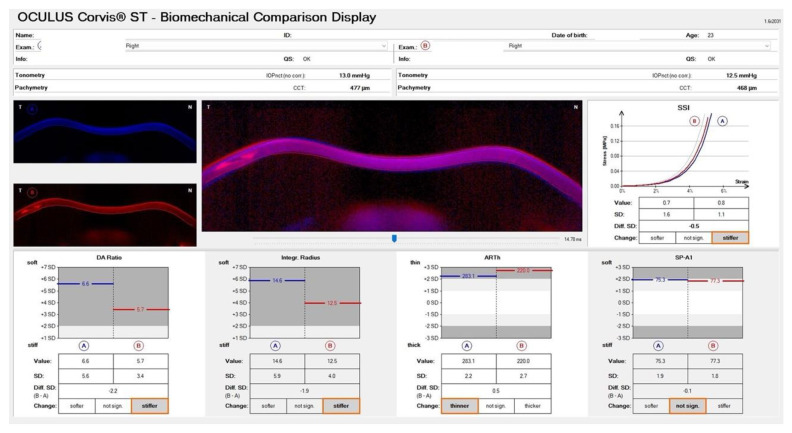
Biomechanical Comparison Display showing a stiffer cornea after the ICRS implantation.

**Table 1 diagnostics-12-03027-t001:** Diagnostic strategies in keratoconus.

Diagnostic Strategies	What Is?	How?
Screening	Detect mild forms of KC, and ectasia susceptibility, considering the refractive treatment and the impact on the cornea.	Placido-disk corneal topography, Scheimpflug tomography, OCT (or VHF US) segmental tomography, and biomechanical assessments.
Diagnostic confirmation	Paradigm shift related to the management of ECD and access ectasia risk and progression to improve treatment.	Comprehensive clinical evaluation with multimodal imaging.
Classification of ectasia	Group of disorders characterized by progressive thinning and following protruding of the corneal structure.	Integration of tomographic and biomechanical data with AI, genetics, and molecular biology.
Staging	To prevent visual loss before it even occurs, with new treatment modalities.	ABCD + E ectasia/KC staging.
Prognostic	Management of KC varies depending on the severity of the disease	Biomechanical parameters (SPA-1) and patient compliance.

Keratoconus (KC), ectatic corneal diseases (ECD), artificial intelligence (AI).

**Table 2 diagnostics-12-03027-t002:** Exams for multimodal propaedeutics in keratoconus.

Imaging Tests	Characterization
Corneal Topography	Analysis of the front surface of the cornea using Placido-disk-based reflection.
Corneal Tomography	Three dimensional reconstruction of the cornea enables the calculation of elevation maps of the front and back surfaces, along with a pachymetric map, typically with rotating Scheimpflug imaging.
Segmental Corneal Tomography	Tomographic evaluation of segments of the cornea, including epithelium, Bowman’s layer, and Descemet’s membrane.
Corvis ST	Non-contact tonometer system that uses an ultra-high-speed Scheimpflug camera to monitor the corneal deformation response over a 5–6 mm area during a constant application of an air pulse, allowing for a more detailed assessment of the deformation process.

## Data Availability

Not applicable.

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
