# Peer review of "Enhanced Diagnostics for Corneal Ectatic Diseases: The Whats, the Whys, and the Hows"

_diagnostics, 2022, doi:10.3390/diagnostics12123027_

Round 1

Reviewer 1 Report

Diagnostics manuscript # diagnostics-2002250, titled "Enhanced Diagnostics for Corneal Ectatic Diseases: The What`s, Why`s, and How`s", is a review article of these authors work diagnosing corneal ectasia including using corneal biomechanical testing and artificial intelligence.  Their work is good, but it needs major editing with some proficient in the English language.  Other minor edits are as follows: 

1. 3rd paragraph, 1st sentence: delete "and" before digital and ocular.  

2. 3rd paragraph, 2nd sentence: delete comma after models.

3. Multimodal Imaging section, 2nd paragraph, last sentence: delete "(not over)".

4. Screening for Ectasia Risk ... section, 1st paragraph, last sentence: What are the several risk factors?

5. Corneal Topography section, 3rd paragraph, 2nd sentence: add ")" after ectasia.

Author Response

We would like to thank your comments and suggestions on our paper. We improved the English and edited the listed five points as suggested.

Reviewer 2 Report

I have read with great interest the manuscript entitled ''Enhanced Diagnostics for Corneal Ectatic Diseases: The What`s, Why`s, and How`s'' where the authors give us an excellent review of the current technology for the diagnosis of ectasias corneal. I really enjoyed reading it a lot and I think it could be published.

Although the authors collect the different cut-off points of the tests to be carried out when diagnosing KC, something more is needed about the demographic data of the KC, risk factors and, above all, I consider that a subsection of childhood Keratoconus could contribute to this nice review

Author Response

We would like to thank your comments and suggestions on our paper.

The article aims to describe more advanced diagnoses for keratoconus. We discuss the risk factors for its possible appearance after refractive surgery, such as the act of eye rubbing, and add other factors to the Screening session, such as young age and low preoperative pachymetry.

In the Follow-up section, we report a case of keratoconus in a 13-year-old patient and its follow-up, including already keratoconus in younger patients.

Reviewer 3 Report

The aim of the article is to present enhanced diagnostics for corneal ectatic diseases.

The English language is appropriate and understandable.

The manuscript is presented in a well structured manner.

References do have autocitations regarding research. Most of the references are within last 5 to 10 years.

Images are appropriate and are easy to interpret.

The aim of the paper is not entirely clear. If the goal is to write a review, then it is necessary to further expand the topic. This text is suitable for a textbook, not as a real scientific review. On the other hand, if the goal of this paper is to present a case report, then it is also deficient because the introduction is too extensive, and the case presentation itself is insufficiently processed.

I think that the editor-in-chief should suggest to the authors, after the revision, the direction of this interesting paper in the direction of a more extensive review or a more concise presentation of the specific case.

Author Response

We want to thank your comments and suggestions on our paper.

The article discusses multimodal propaedeutics and the most current diagnostic indexes for diagnosing and monitoring keratoconus. The reported cases illustrate the importance of these analyzes in clinical practice. The article already discusses the most current topics and predicts the improvement of the diagnosis of the disease with the integration of what was revealed with the use of artificial intelligence.

Round 2

Reviewer 3 Report

No further comments.